# Universality, Robustness, and Detectability of Adversarial Perturbations under Adversarial Training

## Abstract

Classifiers such as deep neural networks have been shown to be vulnerable against adversarial perturbations on problems with high-dimensional input space. While adversarial training improves the robustness of classifiers against such adversarial perturbations, it leaves classifiers sensitive to them on a non-negligible fraction of the inputs. We argue that there are two different kinds of adversarial perturbations: shared perturbations which fool a classifier on many inputs and singular perturbations which only fool the classifier on a small fraction of the data. We find that adversarial training increases the robustness of classifiers against shared perturbations. Moreover, it is particularly effective in removing universal perturbations, which can be seen as an extreme form of shared perturbations. Unfortunately, adversarial training does not consistently increase the robustness against singular perturbations on unseen inputs. However, we find that adversarial training decreases robustness of the remaining perturbations against image transformations such as changes to contrast and brightness or Gaussian blurring. It thus makes successful attacks on the classifier in the physical world less likely. Finally, we show that even singular perturbations can be easily detected and must thus exhibit generalizable patterns even though the perturbations are specific for certain inputs.

## 1 Introduction

While deep learning is relatively robust to random noise (Fawzi et al., 2016), it can be easily fooled by so-called adversarial perturbations (Szegedy et al., 2014). These perturbations are generated by adversarial attacks (Goodfellow et al., 2015; Moosavi-Dezfooli et al., 2016; Carlini & Wagner, 2017b) that generate perturbed versions of the input which are misclassified by a classifier and, at the same time, remain quasi-imperceptible for humans. There have been different approaches for explaining properties of adversarial examples and why they exist in the first place (Goodfellow et al., 2015; Tanay & Griffin, 2016; Fawzi et al., 2017a;b).

Adversarial perturbations often transfer between different network architectures (Szegedy et al., 2014). Moreover, adversarial perturbations have been shown to be relatively robust against various kind of image transformations and can even be successful when being placed as artifacts in the physical world (Kurakin et al., 2017a; Sharif et al., 2016; Evtimov et al., 2017). While adversarial perturbations are data-dependent in that they were generated to fool a classifier on a specific input, there also exist universal perturbations which mislead a classifier on the majority of the inputs (Moosavi-Dezfooli et al., 2017a;b; Metzen et al., 2017b).

Several methods have been proposed for increasing the robustness of deep networks against adversarial examples such as adversarial training (Goodfellow et al., 2015; Kurakin et al., 2017b), virtual adversarial training (Miyato et al., 2017), ensemble adversarial training (Tramèr et al., 2017a), defensive distillation (Papernot et al., 2016; Papernot & McDaniel, 2017), stability training (Zheng et al., 2016), robust optimization (Madry et al., 2017), and Parseval networks (Cisse et al., 2017). An alternative approach for defending against adversarial examples is to detect and reject them as malicious (Metzen et al., 2017a). While some of these approaches actually improve robustness against adversarial examples, the classifier remains vulnerable against adversarial perturbations on a non-negligible fraction of the inputs.

While adversarial training is arguably the most popular approach for increasing the robustness of deep networks, few works have investigated its effect on the resulting classifiers and adversarial perturbations. Recently, Tramèr et al. (2017b) investigated the effect of adversarial training on the classifier's decision boundary and found that it displaces the boundary slightly but not sufficiently for preventing black-box attacks (attacks where the adversarial perturbation is generated based on a source model that is different from the target model). While Tramèr et al. (2017b) focus on the transferability of perturbations in a black-box scenario, we investigate in this work properties of adversarial perturbations under adversarial training in a white-box scenario, where source and target model are identical.

We first study a property of adversarial perturbations which we denote as *sharedness*. We define a perturbation as shared if it fools the classifier on many inputs (at least 2%). We empirically find that adversarial training is effective in removing shared perturbations but does not improve robustness against singular perturbations, which only fool the classifier on a very small subset of the data. Moreover, we study how adversarial training affects the existence of universal perturbations, the robustness of adversarial perturbations against image transformations, and the detectability of adversarial perturbations. In summary, we find that adversarial training is very effective in removing universal perturbations and that the remaining (non-universal, singular) adversarial perturbations are less robust against most transformations. Thus, adversarial training is more promising in preventing certain kind of attacks on system safety, such as those performed in the physical world, than might have been assumed. Moreover, we find that adversarial training leaves the remaining singular adversarial perturbations easily detected. Thus, these perturbations must exhibit some generalizable patterns that can be detected. Understanding and exploiting those patterns might ideally lead to identifying new ways for improving adversarial training.

## 2 BACKGROUND

Let $\xi$ denote an *adversarial perturbation* for an input $\mathbf{x}$ and let $\mathbf{x}^{\text{adv}} = \mathbf{x} + \xi$ denote the corresponding *adversarial example*. The objective of an adversary is to find a perturbation $\xi$ which changes the output of the model in a desired way. For instance, an untargeted attack would search for a perturbation that makes the true or predicted class less likely while a targeted attack would try to make a designated target class more likely. At the same time, the adversary typically tries to keep $\xi$ quasi-imperceptible by, e.g., bounding its $\ell_\infty$-norm. One of the first methods for generating adversarial examples was proposed by Szegedy et al. (2014). While this method was able to generate adversarial examples successfully for many inputs and networks, it was also expensive to compute since it involved an L-BFGS-based optimization.

Goodfellow et al. (2015) proposed a non-iterative and hence fast method for computing adversarial perturbations. This *fast gradient-sign method* (FGSM) defines an adversarial perturbation as the direction in image space which yields the highest increase of a linearization of the cost function $J$ under $\ell_\infty$-norm. This can be achieved by performing one step in the gradient sign's direction with step-width $\varepsilon$: $\xi = \varepsilon \operatorname{sgn}(\nabla_{\mathbf{x}} J(\theta, \mathbf{x}, \mathbf{y}^{\text{true}}))$. Here, $\varepsilon$ is a hyper-parameter governing the distance between original image and adversarial image, $\theta$ denotes the classifier's parameters, and $\mathbf{y}^{\text{true}}$ the true class for input $\mathbf{x}$. As an extension, Kurakin et al. (2017a) introduced the *basic iterative method* (BI), an iterative version of FGSM. They apply FGSM several times with a smaller step size $\alpha$ and clip all pixels after each iteration to ensure results stay in the $\varepsilon$-neighborhood of the original image: $\xi^{(0)} = \mathbf{0}; \xi^{(n+1)} = \operatorname{Clip}_\varepsilon \left\{ \xi^{(n)} + \alpha \operatorname{sgn}(\nabla_{\mathbf{x}} J(\theta, \mathbf{x} + \xi^{(n)}, \mathbf{y}^{\text{true}})) \right\}$.

*Adversarial training* denotes a modification of the training procedure which has been shown to increase robustness of classifiers against adversarial perturbations and at the same acts as a regularizer (Goodfellow et al., 2015). To this end, during adversarial training the objective function $J$ is replaced by $\tilde{J}(\theta, \mathbf{x}, \mathbf{y}^{\text{true}}) = \sigma J(\theta, \mathbf{x}, \mathbf{y}^{\text{true}}) + (1 - \sigma)J(\theta, \mathbf{x} + \xi(x), \mathbf{y}^{\text{true}})$. This can be seen as an online data augmentation, which replaces a fraction of the inputs with their adversarial counterparts. When using this version of adversarial training, Kurakin et al. (2017b) observed label leaking: adversarial perturbations may leak information regarding the true class into the data since their generation is based on the true label. We avoid this by using the class $\mathbf{y}^{\text{pred}}$ predicted by the current classifier for input $\mathbf{x}$ in place of the true class, i.e., we use $\tilde{J}(\theta, \mathbf{x}, \mathbf{y}^{\text{pred}})$ as objective in adversarial training.

## 3 EXPERIMENTS

We perform our experiments on the CIFAR10 dataset (Krizhevsky, 2009) and use a pre-activation Residual Network (He et al., 2016) with 18 layers and 64-128-256-512 feature maps per level as classifier (results for other datasets and classifiers are contained in Appendix H and I). As adversary we use the basic iterative (BI) adversary with $\varepsilon = 4$ and $\alpha = 1$ (relative to inputs with values between 0 and 255) and 10 iterations. We trained the classifier for 200 epochs without adversarial training using SGD and reached a test accuracy of 94.1%; however, accuracy on adversarial examples was merely 0.5%. Thereupon, we performed adversarial training for 250 epochs with $\sigma = 0$. Adversarial training converged to a classifier with roughly 90% accuracy on unchanged test inputs and 62% accuracy on adversarial inputs (see Appendix A). Notably, the accuracy on adversarial examples for the training data gets to 97.7% after 250 epochs. Thus, the main issue preventing adversarial training from reaching higher performance on the adversarial test examples is apparently overfitting to the perturbations of the training data and not an optimization issue or a lack of network capacity. Moreover, we did not found black-box attacks on the model to be more successful than white-box attacks (see Appendix A) as Tramèr et al. (2017a) observed and attributed to gradient masking. We believe this is due to using a stronger, iterative adversary during adversarial training. In the following sections, we take a closer look at what happens during adversarial training.

### 3.1 STABILITY OF ADVERSARIAL PERTURBATIONS

In this section, we study why adversarial training does not increase the robustness of the classifier against adversarial perturbations of certain inputs. Two possible failure cases exist: (a) the model might remain vulnerable against the same perturbation of an input during adversarial training. We denote such adversarial perturbations as *stable* perturbations. (b) The model learns to become robust against an adversarial perturbation but at the same time becomes vulnerable to other perturbations. This might lead to an oscillation-like pattern, where for any perturbation there exist a point in time in which the model is robust against this perturbation, but there does not exist a point in time where the model is robust to all possible perturbations of an input.

For deciding which of the two failure cases is prevalent, we propose to compute a *temporal stability profile* $P$ of a test input under $T$ epochs of adversarial training. This profile $P$ is a $T \times T$ matrix containing the probability of the true class predicted by the classifier after $t_2$ epochs for the adversarial examples generated for the classifier after $t_1$ epochs of adversarial training. More specifically, after $t_1$ epochs of adversarial training, we generate an adversarial perturbation $\xi_{t_1}$ for test input $\mathbf{x}$ and the current model parameters $\theta_{t_1}$. After $t_2$ epochs, we evaluate the predicted probability of the true class $\mathbf{y}$ for the perturbed input under the current model parameters $\theta_{t_2}$ and set the profile to $P_{t_1,t_2}(\mathbf{x}) = p(y|\mathbf{x}+\xi_{t_1};\theta_{t_2})$. For such a profile, we compute both $MaxMin = \min_{t_1} \max_{t_2} P_{t_1,t_2}$ and $MinMax = \max_{t_2} \min_{t_1} P_{t_1,t_2}$. Intuitively, MaxMin being large indicates that for any perturbation of the input, there exists a model during adversarial training that predicts the correct class with high confidence. MaxMin being small thus shows that there exist a stable perturbation of the input that fools all models during adversarial training. MinMax being large, in contrast, indicates that there is a single model which predicts the correct class with high confidence for *all* perturbations of the input. MaxMin being large and MinMax being small thus shows that the model oscillates between being vulnerable to different perturbations of the input.

Figure 1 shows the temporal stability profiles for three different test inputs. The first profile corresponds to an input where adversarial training makes the model robust against all perturbations of the input; the second profile corresponds to a stable perturbation, which fools the model at any point in time (small MaxMin), and the third profile to an oscillation-like pattern. Thus, both stable perturbations and oscillation between perturbations occur. However, we found stable perturbations to be more frequent and being the dominant failure case for adversarial training (see Appendix B).

What causes stable perturbations? We provide evidence that a good predictor of a perturbation being stable is that it is also *singular*. We denote a perturbation as singular if it fools the classifier on a specific input but leaves the classifier's prediction nearly unchanged for most other inputs (98% or more). In contrast, *shared* perturbations are effective for many different inputs. We define the perturbation's cost change on an input $\mathbf{x}$ and parameters $\theta$ as $J^{\Delta}(\xi, \theta, \mathbf{x}, \mathbf{y}) = J(\theta, \mathbf{x} + \xi, \mathbf{y}) - J(\theta, \mathbf{x}, \mathbf{y})$. For a perturbation $\xi_{i,t}$ of an input $\mathbf{x}_i$ and model parameters $\theta_t$, where $t$ indexes the epoch of adversarial training, we can now compute two quantities: (a) how well a perturbation

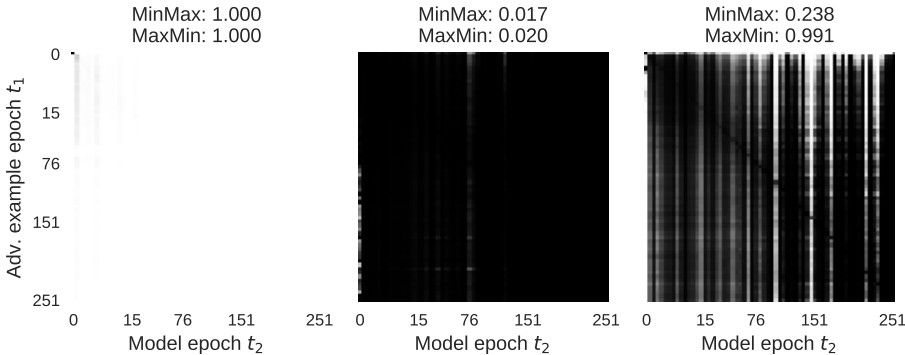

Figure 1: Profiles of three test inputs under adversarial training. White corresponds to a high-confidence correct classification ($P_{t_1,t_2} \approx 1$) and black to an incorrect classification with high confidence ($P_{t_1,t_2} \approx 0$). Note that MaxMin corresponds to first taking the maximum over each row, and then the minimum over the resulting column. MinMax, in contrast, takes first the minimum over the columns and then the maximum over the resulting row.

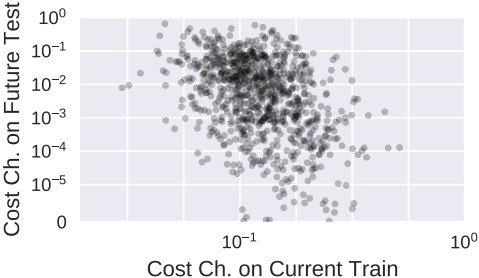

Figure 2: Illustration of the cost change when transferring a perturbation to other inputs from the training set ("sharedness") vs. cost change of a perturbation for future classifier parameters ("stability") under adversarial training for the same input. Please note the log-scales.

transfers to other inputs under the same model parameters $\theta_t$ (its sharedness), and (b) how well a perturbation transfers to other model parameters $\theta_{t_2}$ under the same input (its stability).

More specifically, we randomly sample 1000 inputs $\mathbf{x}_i$ from the test set and generate their respective perturbations $\xi_{i,t=0}$ for the BI adversary and classifier parameters $\theta_{t=0}$, i.e., the classifier before adversarial training. For (a), we randomly sample $N = 10000$ inputs $\mathbf{x}_j$ from the training set and compute $J^\Delta_{\text{current,train}}(\xi_{i,t=0}) = 1/N \sum_j J^\Delta(\xi_{i,t=0}, \theta_{t=0}, \mathbf{x}_j, \mathbf{y}_j)$. For (b), we sample a subset $\mathcal{T}$ of epochs in adversarial training and compute $J^\Delta_{\text{future,test}}(\xi_{i,t=0}) = 1/|\mathcal{T}| \sum_{t \in \mathcal{T}} J^\Delta(\xi_{i,t=0}, \theta_t, \mathbf{x}_i, \mathbf{y}_i)$. Large values of $J^\Delta_{\text{current,train}}$ indicate a shared perturbation since the perturbation increases cost on many inputs while small values indicate a singular perturbation. Similarly, large values under $J^\Delta_{\text{future,test}}$ indicate a stable perturbation.

Figure 2 shows a scatter plot of mean cost-change when transferring a perturbation to an input from the training set (under the same model parameters) versus the cost change when transferring a perturbation to future classifier parameters under adversarial training (for the same input). We note that the two are anti-correlated: the Pearson correlation coefficient in the log-log domain is $-0.293$ ($p < 10^{-18}$). Thus, increased sharedness of a perturbation results in decreased stability: adversarial training makes a classifier robust against shared perturbations which fool the classifier on many inputs but is less effective against singular perturbations. Appendix C shows that the remaining perturbations after adversarial training are in fact singular and rarely transfer between inputs.

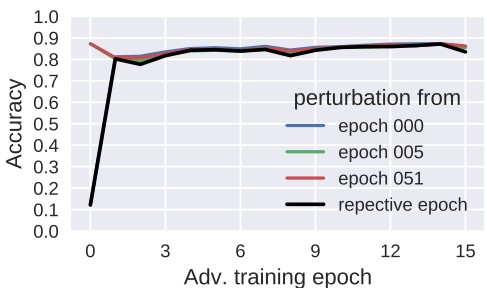

Figure 3: Accuracy under universal perturbations for $\varepsilon = 10$. Shown is the accuracy of the universal perturbation generated for the model from the respective epoch as well as the accuracy of the universal perturbations generated for the model in epoch 0, 5, and 51.

## 3.2 EXISTENCE OF UNIVERSAL PERTURBATIONS

In the previous section, we have empirically shown that adversarial training is effective in removing shared perturbations. An extreme form of shared perturbations are *universal perturbations* (Moosavi-Dezfooli et al., 2017a;b). While shared adversarial perturbations are generated with the objective to fool a classifier on a specific input and are effective for other inputs only as a byproduct, universal perturbations are explicitly generated with the objective of fooling a classifier on as many inputs as possible. Different methods for generating universal perturbations exist: the first approach by Moosavi-Dezfooli et al. (2017a) uses an extension of the DeepFool adversary (Moosavi-Dezfooli et al., 2016) to generate perturbations that fool a classifier on a maximum number of inputs from a training set. Metzen et al. (2017b) proposed a similar extension of the basic iterative method for generating universal perturbations for semantic image segmentation. Mopuri et al. (2017) proposed Fast Feature Fool, a method which is, in contrast to the former works, a data-independent approach for generating universal perturbations. Khrulkov & Oseledets (2017) show a connection between universal perturbations and singular vectors.

To the best of our knowledge, the effectiveness of universal perturbations to fool classifiers trained with adversarial training has not been investigated. Since we have observed in the previous section that adversarial training reduces the sharedness of adversarial perturbations, it appears likely that it is also effective against universal perturbations. Figure 3 confirms this intuition (please refer to Appendix D for details on how we generated universal perturbations): while the model without adversarial training (epoch 0) classifies only 12% of the inputs with the universal perturbation correctly, models trained with adversarial training for 3 or more epochs classify at least 80% of the inputs with universal perturbations correctly. Note that this is true for the universal perturbation generated for the model from the respective epoch as well as for perturbations from other epochs.

Figure 4 illustrates the universal perturbations generated for models in different epochs of adversarial training. From the illustrations, it is apparent that universal perturbations change considerably: while the perturbation for a model without adversarial training (epoch 0) has a clear checker-board like pattern, adversarial training removes this pattern immediately. For the first 15 epochs of adversarial training, the perturbations instead have a black-and-white maze structure at the boundaries. Later on, also this pattern is removed by adversarial training and the universal perturbation consists mostly of large, constant-color areas. This probably corresponds to a situation where perturbations on different training inputs are so different that they effectively cancel out (an indication that they are singular). We would also like to note that also universal perturbations generated on test data are no more effective in fooling the classifier (see Appendix E); it is thus not just an issue of overfitting the perturbations to the training data.

## 3.3 ROBUSTNESS OF ADVERSARIAL PERTURBATIONS

Most work on adversarial examples assumes that the adversary has perfect control of the input of the classifier. This, however, is unrealistic in many scenarios, for instance when the adversary

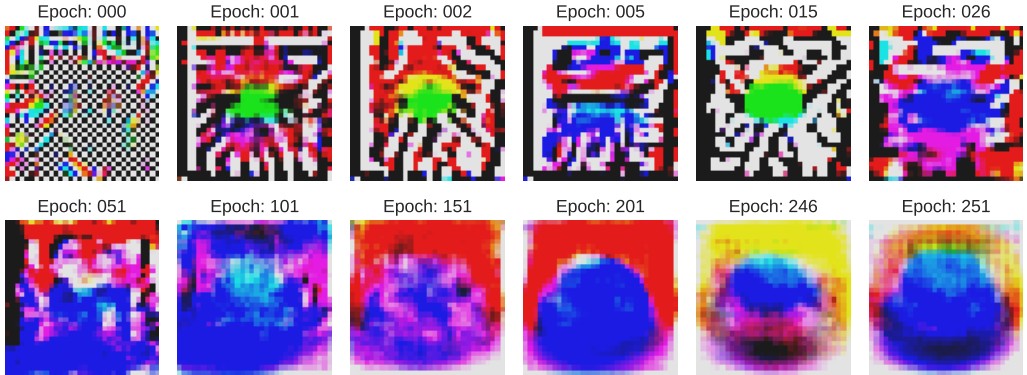

Figure 4: Illustration of universal perturbations for $\varepsilon = 10$ generated for different epochs of adversarial training. The perturbations are amplified by a factor of 10 for readability.

cannot manipulate the digital representation of a scene but only the actual scene in the physical world. Recent work has nevertheless demonstrated that successful attacks on machine learning models under such threat models are possible: Kurakin et al. (2017a) have shown that an adversarial perturbation often remains effective when being printed and recaptured by a camera sensor. Sharif et al. (2016) have demonstrated impersonation and dodging attacks on face recognition systems by adversarial eyeglass frames. Evtimov et al. (2017) have provided evidence that road sign classifiers can be fooled by placing quasi-imperceptible or non-suspicious artifacts on road signs. Finally, Athalye & Sutskever (2017) proposed a method for making adversarial perturbations more robust to transformations such as rescaling, rotation, translation, Gaussian noise, and lightening/darkening.

In this section, we evaluate the effect of adversarial training on the robustness of adversarial perturbations against different transformations. Following Kurakin et al. (2017a), we quantify the robustness of an adversarial example against a transformation via the *destruction rate*. For this let use denote adversarial perturbations which change the classifier's prediction from correct to incorrect as effective. The destruction rate is defined as the fraction of effective adversarial perturbations, which are no longer effective when a transformation is applied. Figure 5 shows the destruction rate of adversarial examples against four different transformations based on Kurakin et al. (2017a) for different models obtained during adversarial training. Clearly, adversarial training increases the destruction rate of adversarial examples under changes of brightness, contrast, and Gaussian blurring. It is notable that a model trained for 251 epochs of adversarial training has a considerably higher destruction rate under these transformations than a model trained for 51 epochs, even though the learning curves (see Appendix A) indicate that there are no improvements on test data for 50 or more epochs. The results for the destruction rate under additive Gaussian noise show a more complex picture: here a model without adversarial training has a smaller destruction rate for weak noise ($\sigma < 5$) but a considerably higher destruction rate for strong noise ($\sigma > 7$) than adversarially trained models. The reason for the latter is unclear and an interesting direction for future work.

In summary, adversarial training not only reduces the number of inputs where the classifier can be fooled but also makes the remaining adversarial examples less robust against typical transformations that would occur in physical world scenarios like brightness changes, contrast changes, or Gaussian blurring. However, there are also transformations like adding strong Gaussian noise where adversarial training appears to be counterproductive. Thus, while it seems likely that adversarial training helps in preventing physical world attacks since those would typically involve changes to brightness and contrast or blurring, further experiments are required for a final conclusion.

## 3.4 DETECTABILITY OF ADVERSARIAL PERTURBATIONS

A different approach for defending against adversarial approaches is trying to identify and reject inputs which have been manipulated by an adversary. Many approaches have been proposed recently for detection of adversarial perturbations; we refer to Carlini & Wagner (2017a) for an overview of

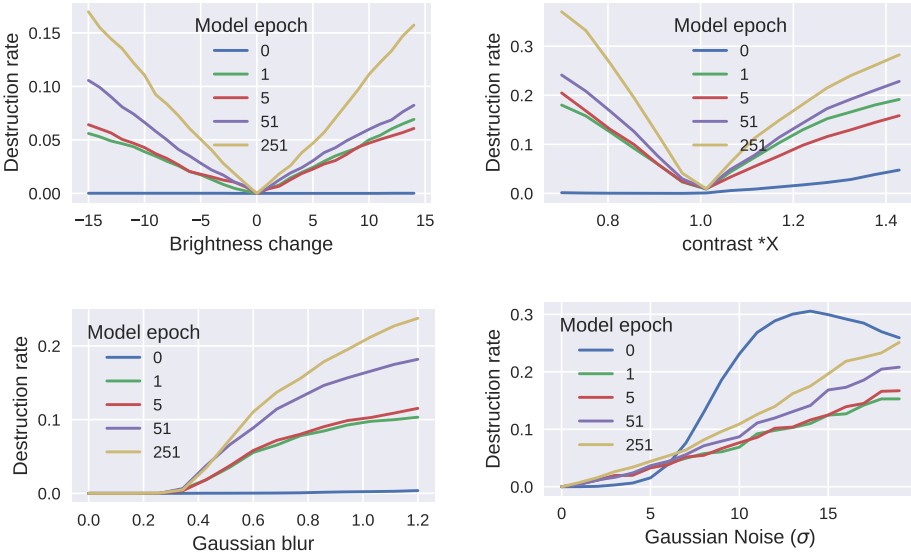

Figure 5: Destruction rate of adversarial perturbations for 4 kind of transformations: additive brightness changes, multiplicative contrast changes, Gaussian blurring, and additive Gaussian noise. Results are for basic iterative adversary, results for other adversaries can be found in Appendix F.

approaches and a cautionary note that detectors can themselves be fooled by an adversary which is aware of their existence. In this section, however, we are not primarily interested in whether detection of adversarial perturbation is an effective defense but rather whether adversarial perturbations are sufficiently regular for being detectable in principle. While this has been answered affirmatively in recent works for scenarios where the adversary does not actively try to fool the detector, it has not been investigated whether adversarial trained models exhibit the same property.

We focus on the approach for detecting adversarial perturbations proposed by Metzen et al. (2017a): in this approach a separate convolutional neural network denoted as *detector* is trained on the binary classification task of distinguishing unchanged from adversarial inputs. We follow the network architecture proposed by Metzen et al. (2017a) and apply it directly to the inputs (and not to intermediate feature maps of the classifier). Figure 6 shows the detectability of adversarial perturbations during adversarial training. It can be seen that the detectability of adversarial perturbation, i.e., the accuracy of the detector, is well above 90% for any time during adversarial training. This indicates that even though adversarial perturbations become increasingly more singular during adversarial training, they must still exhibit generalizable patterns that make them detectable.

Moreover, also the detectors trained on the adversarial examples from epoch 1 and 251 of adversarial training detect adversarial examples generated at any other time during adversarial training with more than 90% accuracy. This indicates that the patterns that make adversarial examples detectable must stay relatively robust over time. This is in contrast to the universal perturbation, which change drastically over time. We would also like to note that a detector trained on the base model that was not adversarially trained (epoch 0) does not reliably detect adversarial perturbations in later epochs. Thus there is a principled change in the adversarial examples in the first epoch of adversarial training.

To get an understanding of the patterns which a detector exploits for identifying adversarial example, we provide an illustration of those patterns in Figure 7. More specifically, we adapted the method for generating universal perturbations from Appendix D such that it does not maximize the classifier's cost but maximizes the detector's predicted probability of the inputs being adversarial. For a model without adversarial training (epoch 0), the generated perturbation shows the same checker-board like pattern as the universal perturbation in Figure 4. For epoch 1 or later, the generated perturbation look already similar to the universal perturbations of late epochs. These perturbations, however, also have a mean probability of being detected as adversarial of 12% or less when added to inputs. Since

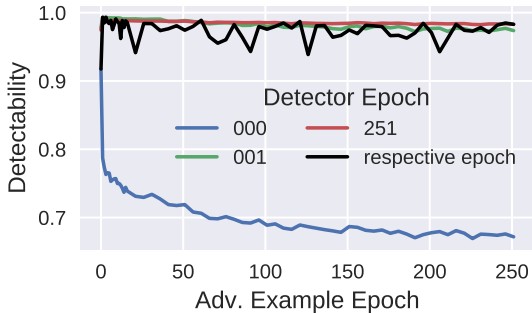

Figure 6: Detectability of adversarial perturbations on test data. Shown is the performance of detectors trained on the respective epoch as well as the performance of detectors trained in fixed epochs. Results are for basic iterative adversary, results for other adversaries can be found in Appendix G.

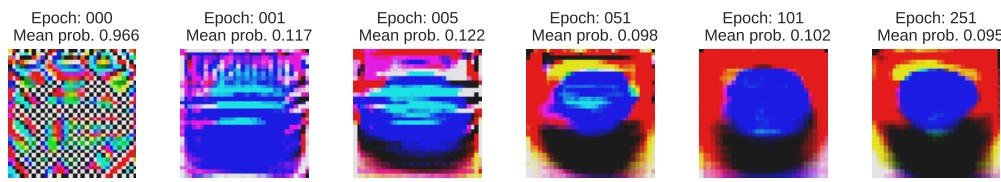

Figure 7: Illustration of perturbations that maximally activate a detector on the training data (for $\varepsilon = 10$). The perturbations are amplified by a factor of 10 for readability. Shown is also the mean probability of an input with the perturbation being adversarial predicted by the detector.

the detector identifies adversarial perturbations with high accuracy in general, we conclude that the detector does not pick up a global, linear pattern but rather input-dependent patterns (corresponding to the singular perturbations), which cancel out in Figure 7.

## 3.5 GENERALITY OF RESULTS

The results presented thus far have been for a specific classifier (a residual network) on a specific dataset (CIFAR10) against a specific adversary (basic iterative gradient sign method). How general are the results, that is: do they generalize to different classifiers, datasets, and adversaries? While we leave the details to the appendix, we would like to briefly summarize the main findings here: Appendix F shows that adversarial training also increases the destruction rate of perturbations generated by other adversaries (FGSM and DeepFool) under all image transformations except for adding Gaussian noise. Appendix G presents results that perturbations generated by FGSM and DeepFool are also well detectable at any time during adversarial training. Appendix H shows results for the same classifier on a different dataset (German Traffic Sign Recognition Benchmark (Stallkamp et al., 2012)) and Appendix I shows results for a non-residual convolutional classifier on this dataset. While specific details differ, the main findings remain valid for different datasets and for different classifiers: adversarial training makes classifiers considerably more robust against universal perturbations, increases the destruction rate of perturbations considerably under most image transformations, and leaves perturbations well detectable.

## 4 DISCUSSION AND CONCLUSION

We have empirically investigated the effect of adversarial training on adversarial perturbations. We have found that adversarial training increases robustness against shared perturbations and even more against universal perturbations. A non-negligible part of the inputs, however, exhibit singular perturbations, which are effective only for the specific input. While adversarial training is not successful

in removing singular perturbations, we found that it makes the remaining singular perturbations less robust against image transformations such as changes to brightness, contrast, or Gaussian blurring. Because of the classifier's reduced vulnerability against universal perturbations and the reduced robustness of the remaining singular perturbations, adversarial training appears very promising for preventing attacks on system safety such as physical-world attacks presented for face recognition (Sharif et al., 2016) or road sign classification (Evtimov et al., 2017). We strongly recommend that future work in those directions investigates the feasibility of attacks not only against undefended classifiers but also against classifiers that have undergone adversarial training or other defense mechanisms since our results indicate that adversarial training might affect the feasibility of such attacks severely.

Interestingly, while singular perturbations are very specific for certain inputs, we found them nevertheless to be sufficiently regular for being detectable. While detection of adversarial perturbations is not yet an effective defense mechanism (Carlini & Wagner, 2017a), our results on detectability indicate that there seem to be stable patterns in adversarial examples but that adversarial training fails to make the classifier itself robust against those patterns. Understanding and using what makes up stable patterns in adversarial examples may be a promising direction for improving adversarial training and increasing robustness against safety-relevant attacks in the physical world. We leave a closer investigation of this to future work.

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

## A    LEARNING CURVES

Learning curves of adversarial training are shown in Figure 8: after an initial drop, the accuracy on unchanged test inputs reaches approximately 90%. The accuracy on adversarial inputs reaches approximately 68% for FGSM and approximately 62% for BI. Notably, the accuracy on adversarial examples for the training data gets close to 100%. The learning curves indicate that after approximately 25 epochs of adversarial training the classifier starts to overfit to particular perturbations of the training set. In the light of our results from Section 3.1, we hypothesize that the classifier learns to become robust against shared perturbations in the first epochs and thereupon (starting after approximately 25 epochs) starts to memorize singular perturbations on the training data. While increased robustness against shared perturbations generalizes to test data, memorizing singular perturbations obviously does not generalize.

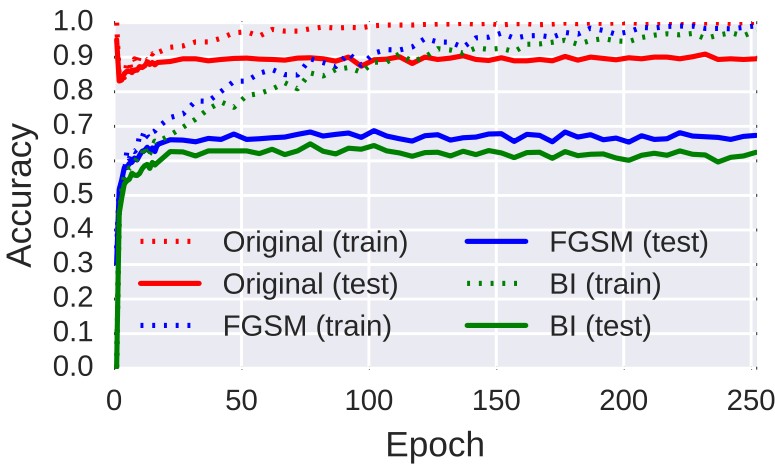

Figure 8: Learning curve during adversarial training.

Figure 9 evaluates the error rate on training and test data for the BI adversary in white-box and different black-box attack scenarios. While the white-box attack is analogous to the results shown in Figure 8 (small differences are due to subsampling the test data to 256 inputs), the black-box attacks are performed as follows: adversarial examples for the classifiers obtained after 0, 5, 15, 51, and 251 epochs of adversarial training (the source models) are generated and tested for all classifiers obtained during adversarial training (the target models). Since typically source and target model are different, the attack is a black-box attack which relies on transferability between models. Tramèr et al. (2017a) observed that black-box attacks can be more effective than white-box attacks for adversarially trained models. They attributed this to gradient masking, i.e., adversarial training not making classifiers more robust but just adversarial examples harder to find with gradient-based approaches. However, we do not observe this effect on either training or test data. We believe this is due to using a stronger, iterative adversary during adversarial training.

We further note that on test data, adversarial examples generated for a classifier after 5 or 15 epochs of adversarial training remain effective against all later classifiers, with an error rate of 20% or more. This further indicates that there exist stable perturbations against which the classifier never becomes robust.

## B    MAXMIN VERSUS MINMAX SCATTER PLOT

As discussed in Section 3.1, we can generate a temporal stability profile of adversarial training for each input. We can summarize this profile in two quantities: MaxMin and MinMax. Figure 10 shows a scatter plot for these two quantities for 256 test inputs. Points being in the upper right corner denote inputs for which adversarial training made the classifier robust against all perturbations. Points in the

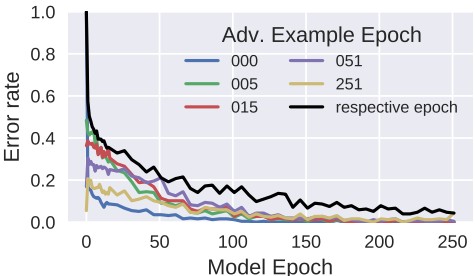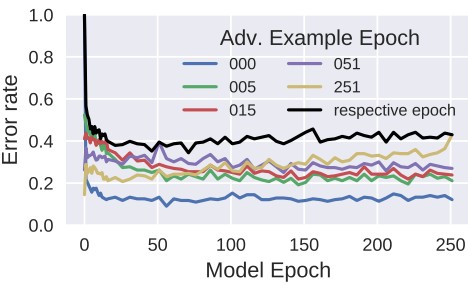

Figure 9: Error rate of white-box and black-box attacks using BI on the classifier during adversarial training on training data (left) and test data (right). "Respective epoch" denotes a white-box attack where the perturbation is generated on the model which is also the target model to be fooled. The other settings denote black-box attacks, where the perturbations are generated on an other model learned during the course of adversarial training. The error rate is evaluated on the same 256 inputs sampled randomly from the test data.

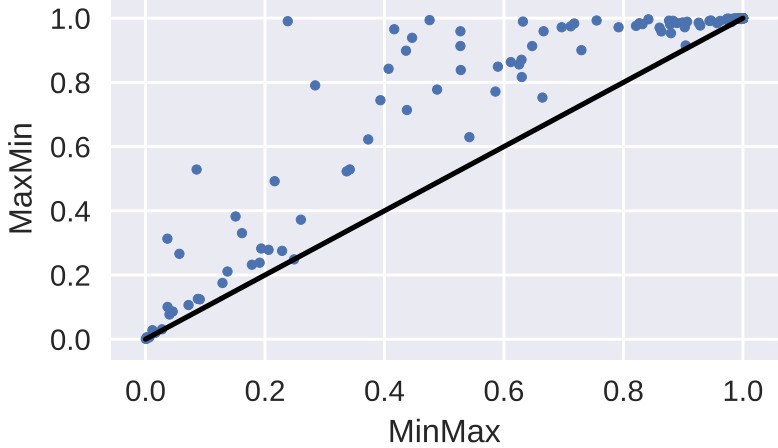

Figure 10: Scatter plot of MinMax versus MaxMin on 256 test inputs.

lower left corner denote inputs where a stable perturbation exists, which fools all classifiers during adversarial training. Points in the upper left corner denote inputs for which the classifier oscillates and is robust against any perturbation at some time but never against all at the same time.

One can see that most points are relatively close to the diagonal, which indicates that there are no oscillations between different perturbations. Points in the lower left corner are the dominant failure case of adversarial training: a stable perturbation which fools the classifier at any time during adversarial training. There are fewer points close to the upper left corner, which correspond to the second failure case: oscillations between different perturbations, where for any perturbation, the classifier is robust against it at one point during adversarial training, but there is no classifier which is robust against all.

## C SHAREDNESS RATIO

In this section, we give quantitative numbers of the sharedness of adversarial perturbations on test data at different epochs during adversarial training. More specifically, we sample 1024 test inputs

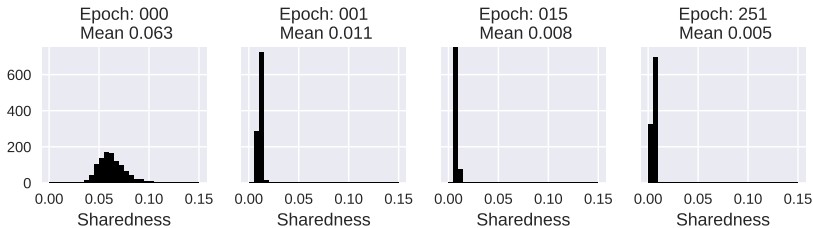

Figure 11: Sharedness of adversarial perturbations on test data during adversarial training.

uniform at random and generate their respective adversarial perturbations for the classifiers after 0, 1, 15, and 251 epochs of adversarial training. We then transfer these perturbations to 5000 other test inputs and compute the percentage of these inputs for which the perturbations are effective, i.e., change the prediction of the respective classifier (from the same epoch) from correct to incorrect.

The results are summarized in Figure 11. Perturbations for the classifier without adversarial training (epoch 0), are shared: on average, they are also effective for 6.3% of the other inputs and at least for 3.3%. Already after 1 epoch of adversarial training, the sharedness is drastically reduced: on average, the perturbations are effective for 1.1% of the other inputs. This is further reduced to 0.8% after 15 epochs and 0.5% after 251 epochs. Notably, the most shared perturbation after 251 epochs is effective for less than 1% of the other inputs. In summary, adversarial training reduces sharedness considerably and most remaining perturbations can be considered as singular perturbations in the sense that they work only for a specific input or a small fraction of very similar inputs (less than 2%).

## D  GENERATING UNIVERSAL PERTURBATIONS

We roughly follow the approach proposed by Metzen et al. (2017b) for generating universal perturbation. While they extend the basic iterative procedure to generating targeted universal perturbations for semantic image segmentation, we adapt the procedure to generating untargeted universal perturbations for image classification. Let $\Xi$ denote the universal perturbation. We generate the universal perturbations as follows:

$$\Xi^{(0)} = \mathbf{0},$$

$$\Xi^{(n+1)} = \mathrm{Clip}_\varepsilon \left\{ \Xi^{(n)} + \frac{\alpha}{K} \sum_{k=1}^{K} \mathrm{sgn}(\nabla^{\mathcal{D}^{(k)}}(\Xi^{(n)})) \right\},$$

with $\nabla^{\mathcal{D}}(\Xi) = \frac{1}{m} \sum_{j=1}^{m} \nabla_x J(\theta, \mathbf{x}_j + \Xi, \mathbf{y}_j^{\mathrm{true}})$ being the loss gradient averaged over a mini-batch of inputs sampled uniform randomly from the training set. In the experiments, we use $\epsilon = 10$, $\alpha = 2.5$, a mini-batch size of $m = 128$, $K = 10$ batches per update, and 200 iterations.

## E  UNIVERSAL PERTURBATIONS ON TEST DATA

Figure 12 shows the accuracy of the classifier during adversarial training against universal perturbations generated on train and test data. Surprisingly, universal perturbations generated on the test data are no more effective than those generated on training data. This indicates that the classifier is actually more robust against universal perturbations and it is not just harder to generate universal perturbations on the training set that generalize to unseen data.

Figure 13 shows illustrations of universal perturbations generated on the test data. For the first epochs, the universal perturbations look visually similar to the perturbations generated on the training data. For later epochs, the test set perturbations are more noisy than the training set perturbations.

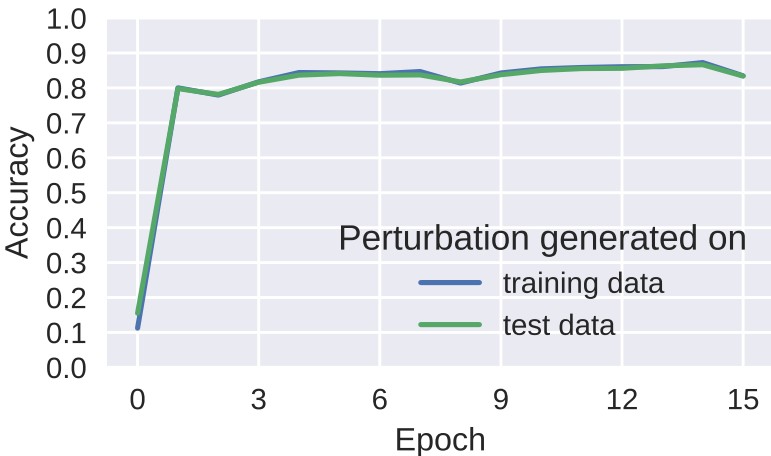

Figure 12: Accuracy on test data for universal perturbations with $\epsilon = 10$ for perturbations generated on training and test data. Note that both lines are nearly identical.

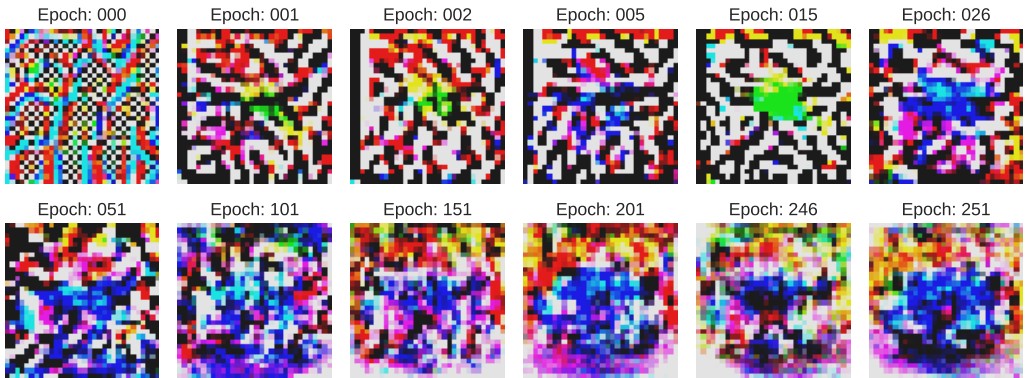

Figure 13: Illustration of universal perturbations for $\varepsilon = 10$ generated on test data. The perturbations are amplified by a factor of 10 for readability.

We suspect this to be caused by the remaining singular perturbations on the test data, which are no longer existent on the training data.

## F   DESTRUCTION RATE FOR DIFFERENT ADVERSARIES

In Section 3.3 we have presented results which indicate that adversarial training leaves the remaining adversarial perturbations less robust to certain transformations of the input such as changes of brightness and contrast or blussring and thus transferrable to the physical world. Those results were specific for the basic iterative adversary; Figure 14 and Figure 15 show the corresponding results for FGSM and DeepFool (Moosavi-Dezfooli et al., 2016). The overall trend is the same: adversarial training increases destruction rate of perturbations for changes in brighness and contrast and gaussian blurring. For additive Gaussian noise, however, perturbations become even more robust than for an undefended model (smaller destruction rate).

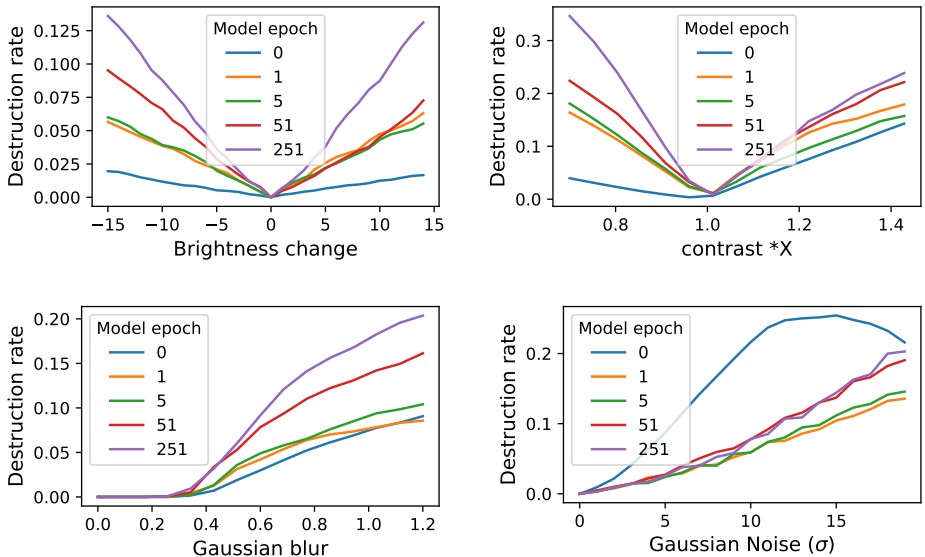

Figure 14: Destruction rate of adversarial perturbations of FGSM adversary for 4 kind of transformations: additive brightness changes, multiplicative contrast changes, Gaussian blurring, and additive Gaussian noise.

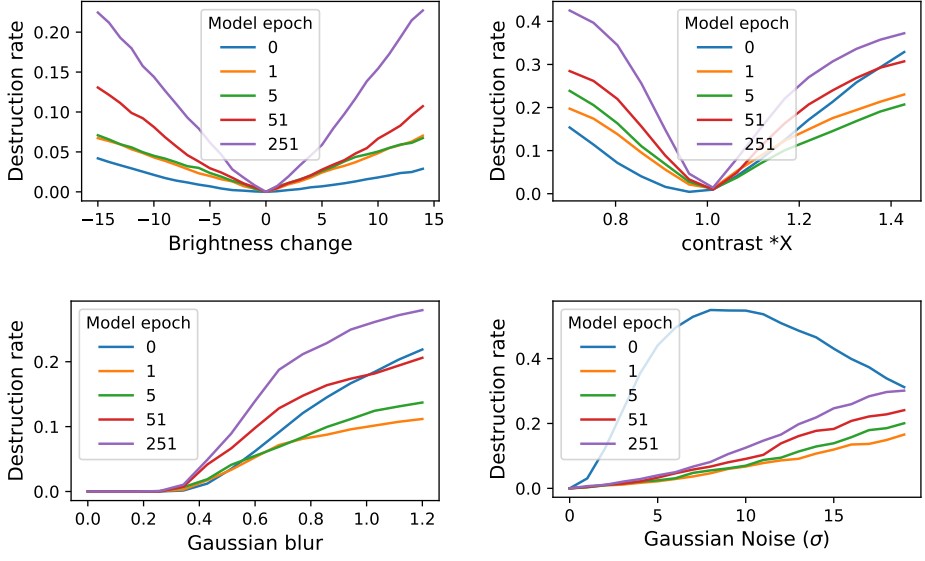

Figure 15: Destruction rate of adversarial perturbations of DeepFool adversary for 4 kind of transformations: additive brightness changes, multiplicative contrast changes, Gaussian blurring, and additive Gaussian noise.

## G    DETECTABILITY OF DIFFERENT ADVERSARIES

Figure 16 shows the detectability (compare Section 3.4) for the FGSM and the DeepFool adversaries. As has been noted by Metzen et al. (2017a), in general FGSM is easier to detect and DeepFool is harder to detect than the basic iterative adversary. Besides that, the results reinforce the main finding

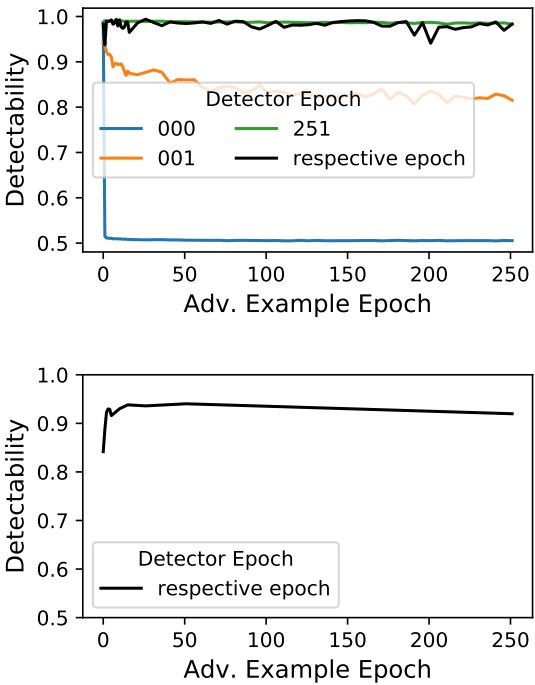

Figure 16: Detectability of adversarial perturbations on test data. Shown is the performance of detectors trained on the respective epoch as well as the performance of detectors trained in fixed epochs. Upper plot: Detectability of FGSM. Bottom plot: Detectability of DeepFool. The evaluation of the detectability of DeepFool has been restricted to the "respective epoch" setting because of the large computational cost of computing the respective perturbations.

of Section 3.4: adversarial perturbations remain detectable even when shared perturbations have been removed by adversarial training.

## H  GERMAN TRAFFIC SIGN RECOGNITION BENCHMARK

We repeat our experiments from Section 3 on the German Traffic Sign Recognition Benchmark (GTSRB) (Stallkamp et al., 2012). Since we found that in general, robustness against adversarial perturbations is larger on GTSRB than on CIFAR10, we use $\varepsilon = 8$ during adversarial training and for generating image-dependent perturbations. We split the GTSRB training set in 35000 samples used for training the network and the remaining 4209 samples for evaluation. We do not use a pretrained classifier (in contrast to our experiments in Section 3) but use the first 25 epochs for vanilla, non-adversarial training. Thereupon, we perform 175 epochs of adversarial training. Otherwise, we use the same hyperparameters as in Section 3.

Figure 17 shows the learning curve during adversarial training. Despite the larger value of $\varepsilon$, accuracy under adversarial perturbations generated using BI reaches 95.2% on test data and 100% on training data. This shows that it is considerably easier to reach high levels of robustness on GTSRB than on CIFAR10. Moreover, nearly perfect robustness can be achieved on training data (as on CIFAR10).

Figure 18 shows the accuracy of the classifier for universal perturbations of $\varepsilon = 20$ (please note that $\varepsilon$ has been doubled compared to the experiments on CIFAR10). Adversarial training is again effective in increasing the robustness of a classifier against universal perturbations: accuracy on data containing universal perturbations is increased from approx. 40% to more than 98%. Figure 19 illustrates the corresponding universal perturbations: universal perturbations for classifiers without

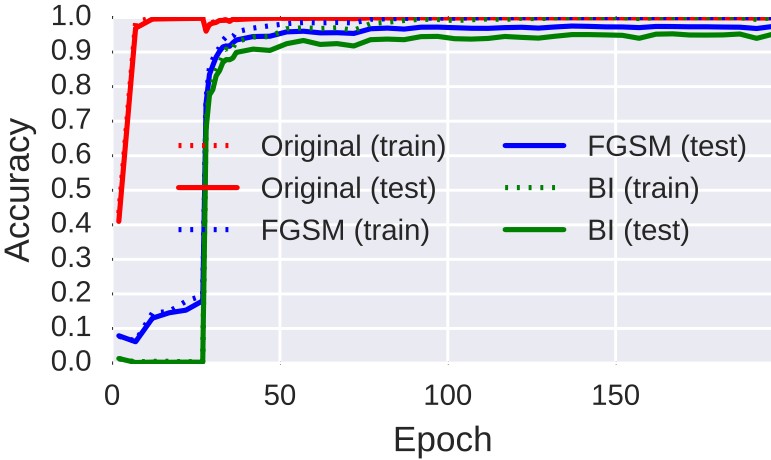

Figure 17: Learning curve on GTSRB for a residual net. First 25 epochs are vanilla training, thereupon we perform adversarial training with $\varepsilon = 8$.

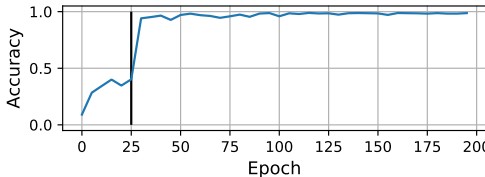

Figure 18: Accuracy under universal perturbations for $\varepsilon = 20$ on GTSRB. Shown is the accuracy of the universal perturbation generated for the model from the respective epoch.

adversarial training correspond to high-frequency noise. Adversarial training immediately removes these kinds of universal perturbations (and increases robustness against those considerably).

Figure 20 shows the destruction rate of adversarial perturbations under different image transformations on GTSRB. In summary, adversarial training increase the destruction rate under changes of brightness, reduction of contrast, and adding Gaussian noise. For Gaussian blur, adversarial training increases the destruction rate for small amount of blurring but reduces it slightly for strong blurring.

Figure 21 shows the detectability of adversarial perturbations on GTSRB. Since $\varepsilon$ had to be increased from 4 to 8 to achieve a sufficient fooling rate on GTSRB, the detectability is also increased considerably. For a detector trained on the respective episode, it remains close to 100% and also detectors trained on epoch 26 and on episode 195 (model after one epoch and at the end of adversarial training) transfer with nearly perfect detectability to models from different epochs. This shows that there is little qualitative change in adversarial perturbation during training on GTSRB, the change is mostly that many perturbations no longer fool the model.

## I  OTHER CLASSIFIER ARCHITECTURE

We repeat our experiments on GTSRB (Appendix H) for a non-residual, convolutional neural net. More specifically, we use a convolutional net with the architecture C32-C32-MP-D0.2-C64-C64-MP-D0.2-C128-C128-MP-D0.2-L512-D0.5-L43, where C32 denotes a convolutional layer with 32 feature maps, MP denotes 2x2 max pooling, D0.2 denotes dropout with p=0.2, and L512 denotes a dense layer with 512 units. ReLU is used as non-linearity throughout the network.

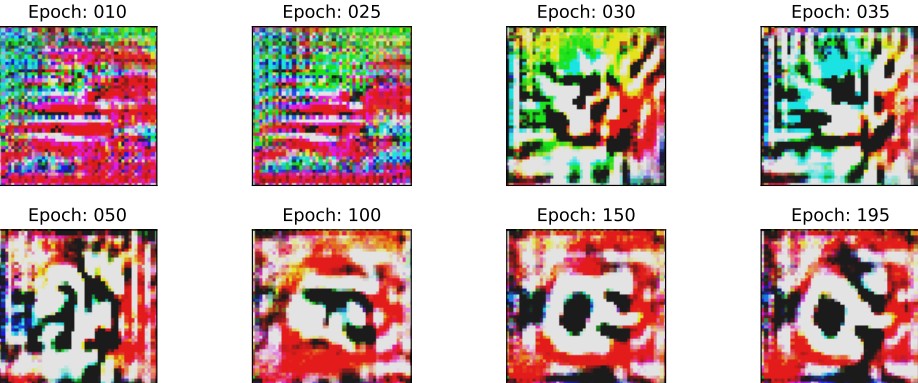

Figure 19: Illustration of universal perturbations for $\varepsilon = 20$ generated for different epochs of adversarial training. The perturbations are amplified by a factor of 5 for readability.

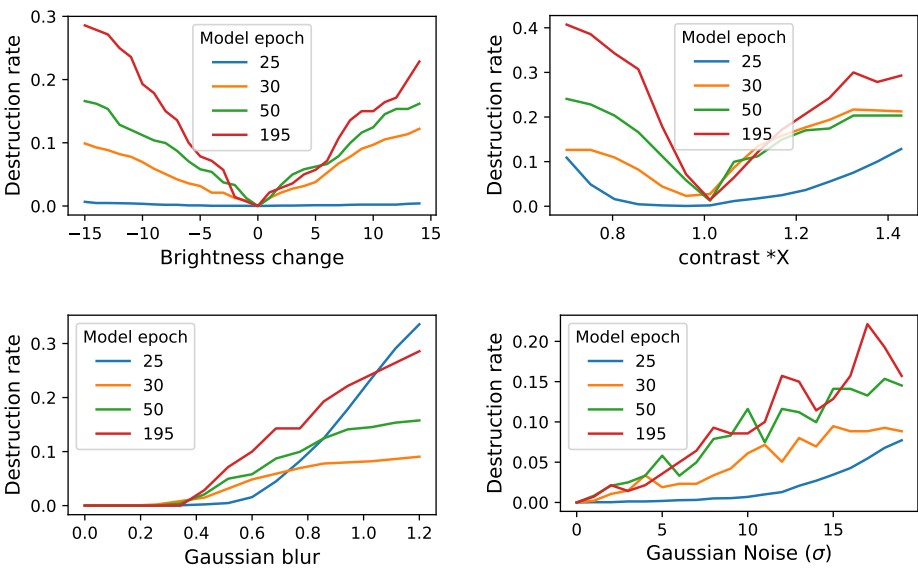

Figure 20: Destruction rate of adversarial perturbations for 4 kind of transformations on GTSRB: additive brightness changes, multiplicative contrast changes, Gaussian blurring, and additive Gaussian noise. Results are for basic iterative adversary.

Figure 22 shows the learning curve during adversarial training. Despite the larger value of $\varepsilon$ compared to CIFAR10, accuracy under adversarial perturbations generated using BI reaches 90% on test data and 97% on training data. Interestingly, the classifiers before adversarial training is turned on are more robust against adversarial perturbations than for the residual network. However, ultimately, the residual network becomes more robust than the non-residual after sufficient amount of adversarial training (see Figure 17).

Figure 23 shows the accuracy of the convolutional classifier for universal perturbations of $\varepsilon = 20$ (please note that $\varepsilon$ has been doubled compared to the experiments on CIFAR10). Adversarial training is again effective in increasing the robustness of a classifier against universal perturbations: accuracy on data containing universal perturbations is increased from approx. 60% to more than 95%. Figure 24 illustrates the corresponding universal perturbations: in contrast to the results of a residual network, there is no qualitative change in the universal perturbation (from the very be-

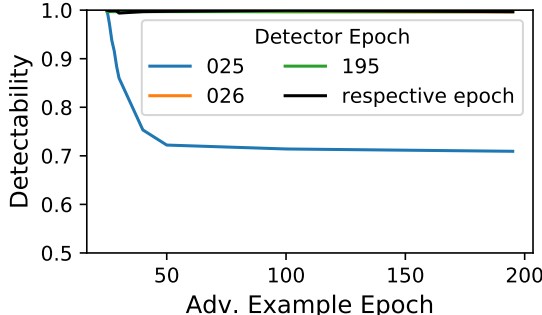

Figure 21: Detectability of adversarial perturbations on GTSRB. Shown is the performance of detectors trained on the respective epoch as well as the performance of detectors trained in fixed epochs. Results are for basic iterative adversary, results for other adversaries can be found in Appendix G.

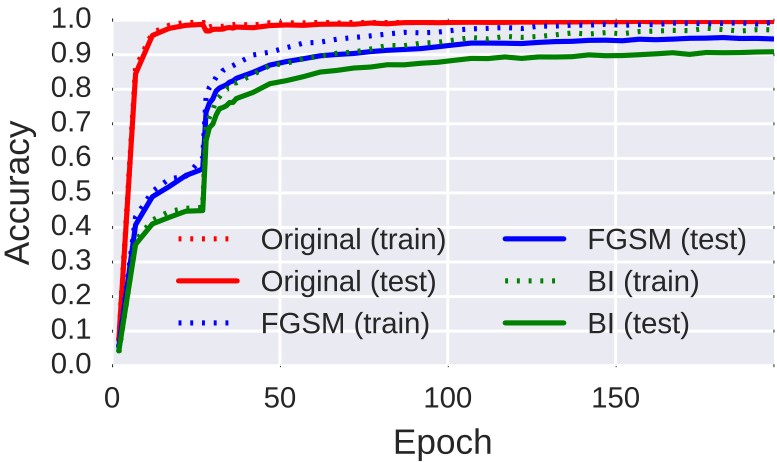

Figure 22: Learning curve on GTSRB for a convolutional, non-residual net. First 25 epochs are vanilla training, thereupon we perform adversarial training with $\varepsilon = 8$.

ginning, they consist mostly of black or white curvy segments and no high-frequency noise pattern exists). Nevertheless, adversarial training greatly reduces their detrimental effect on accuracy.

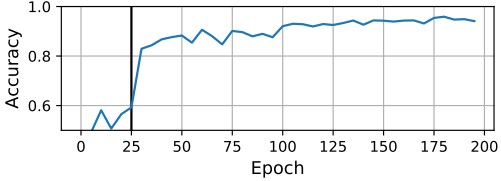

Figure 23: Accuracy under universal perturbations for $\varepsilon = 20$ on GTSRB for a convolutional, non-residual net. Shown is the accuracy of the universal perturbation generated for the model from the respective epoch.

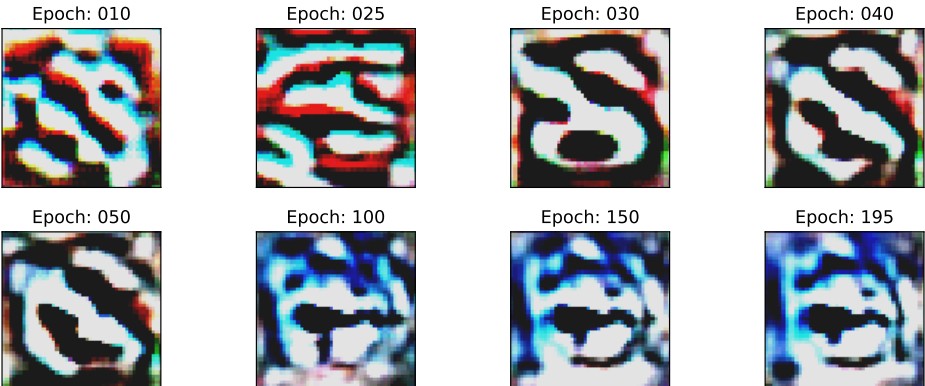

Figure 24: Illustration of universal perturbations for $\varepsilon = 20$ generated for different epochs of adversarial training for a convolutional, non-residual net. The perturbations are amplified by a factor of 5 for readability.

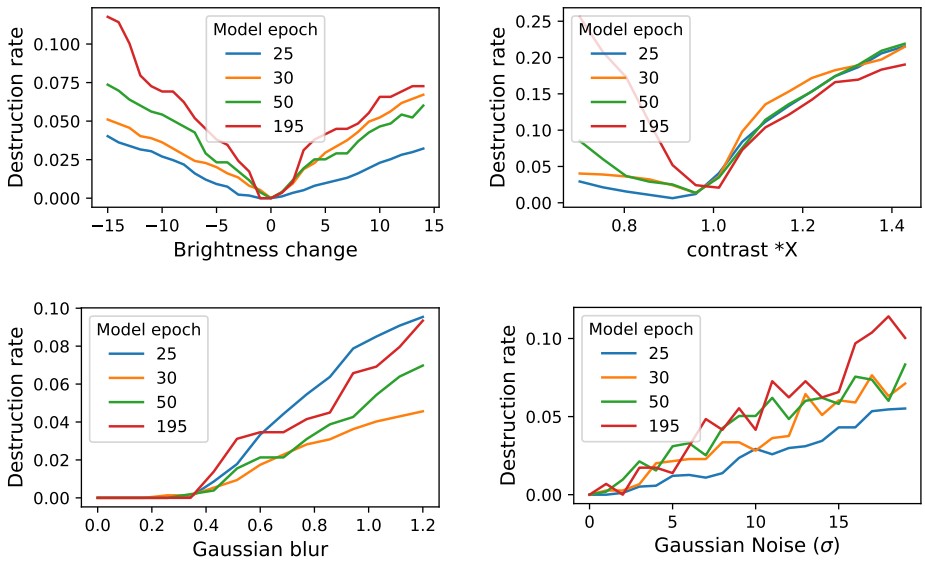

Figure 25: Destruction rate of adversarial perturbations for 4 kind of transformations on GTSRB for a convolutional, non-residual net: additive brightness changes, multiplicative contrast changes, Gaussian blurring, and additive Gaussian noise. Results are for basic iterative adversary.

Figure 25 shows the destruction rate of adversarial perturbations under different image transformations on GTSRB for the non-residual network. In summary, adversarial training increases the destruction rate under changes of brightness, reduction of contrast, and adding Gaussian noise. For Gaussian blur, adversarial training initially reduces the destruction rate (it leaves less perturbations adversarial but those are more robust to image transformations); however, in the long run, the destruction rate reaches a similar level as before.

Figure 26 shows the detectability of adversarial perturbations on GTSRB for the non-residual classifier. Since $\varepsilon$ had to be increased from 4 to 8 to achieve a sufficient fooling rate on GTSRB, the detectability is also increased considerably. For a detector trained on the respective episode, it remains close to 100% and also detectors trained on epoch 25 (model before adversarial training) and on episode 195 (model at the end of adversarial training) transfer with nearly perfect detectability to

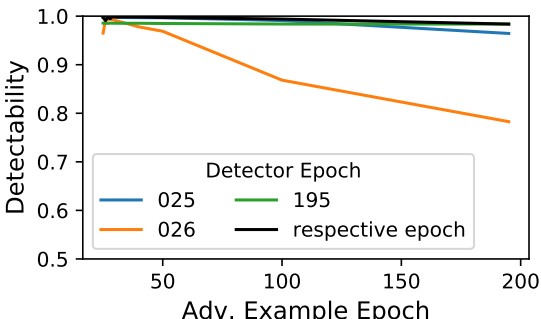

Figure 26: Detectability of adversarial perturbations on GTSRB for a convolutional, non-residual net. Shown is the performance of detectors trained on the respective epoch as well as the performance of detectors trained in fixed epochs. Results are for basic iterative adversary, results for other adversaries can be found in Appendix G.

models from different epochs. This shows that there little qualitative change in adversarial perturbation during training on GTSRB, the change is mostly that many perturbations no longer fool the model.

