# OpenReview forum: "Universality, Robustness, and Detectability of Adversarial Perturbations under Adversarial Training"
_ICLR.cc/2018/Conference — Reject_

### Official Review · AnonReviewer3 · 2017-11-27
**An interesting area of research, but paper lacks focus, does not give new theoretical insight and provides limited experiments.**

**Rating:** 3
**Confidence:** 4

**Review:**

Summary:

This paper empirically studies adversarial perturbations dx and what the effects are of adversarial training (AT) with respect to shared (dx fools for many x) and singular (only for a single x) perturbations. Experiments use a (previously published) iterative fast-gradient-sign-method and use a Resnet on CIFAR.

The authors conclude that in this experimental setting:
- AT seems to defend models against shared dx's.
- This is visible on universal perturbations, which become less effective as more AT is applied.
- AT decreases the effectiveness of adversarial perturbations, e.g. AT decreases the number of adversarial perturbations that fool both an input x and x with e.g. a contrast change.
- Singular perturbations are easily detected by a detector model, as such perturbations don't change much when applying AT.

Pro:
- Paper addresses an important problem: qualitative / quantitative understanding of the behavior of adversarial perturbations is still lacking.
- The visualizations of universal perturbations as they change during AT are nice.
- The basic observation wrt the behavior of AT is clearly communicated.

Con:
- The experiments performed are interesting directions, although unfocused and rather limited in scope. For instance, does the same phenomenon happen for different datasets? Different models?
- What happens when we use adversarial attacks different from FGSM? Do we get similar results?
- The papers lacks a more in-depth theoretical analysis. Is there a principled reason AT+FGSM defends against universal perturbations?

Overall:
- As is, it seems to me the paper lacks a significant central message (due to limited and unfocused experiments) or significant new theoretical insight into the effect of AT. A number of questions addressed are interesting starting points towards a deeper understanding of *how* the observations can be explained and more rigorous empirical investigations.

Detailed:
-

---

> ### Author Response · Authors · 2017-12-20
> **Reply to AnonReviewer3**
>
> We would like to thank the reviewer for the comments. Regarding the reviewer's questions:
> * "The experiments performed are interesting directions, although unfocused". In our opinion, the paper is focused on the question "how does adversarial training affect properties of adversarial examples?" We study a number of properties of adversarial examples such as their sharedness, universality, detectability, and robustness against image transformations. We think it is important to consider all these properties together rather than purely focusing on a single property such as the fooling rate of the model. We tried to connect the different experiments with a common thread.
> * "Does the same phenomenon happen for different datasets? Different models?"
> We have added experimental results on the German Traffic Sign recognition Benchmark (GTSRB) to the revised version of  paper. We have also evaluated a different model (a non-residual, convolutional network) on GTSRB. The main findings remain the same: adversarial training makes classifiers considerably more robust against universal perturbations, increases the destruction rate of perturbations considerably under most image transformations, and leaves perturbations well detectable.
> * "What happens when we use adversarial attacks different from FGSM? Do we get similar results?" Results reported in the paper were for the Basic Iterative (BI) adversary. We have added results for FGSM and DeepFool to the appendix of the revised version of the paper (both regarding destruction rate and detectability). The main findings remain the same for these adversaries.
> * "The papers lacks a more in-depth theoretical analysis. Is there a principled reason AT+FGSM defends against universal perturbations?" Our main results are empirical; however, we think that they provide potential direction for future theoretical analysis and are useful for this reason. It would be, for instance, interesting to connect our empirical findings on universal perturbations to the theoretical insights from the paper "Analysis of universal adversarial perturbations" (https://infoscience.epfl.ch/record/228329/files/universal_perturbations_theory.pdf). In this paper, the authors  show that (assuming a locally curved decision boundary model) the existence of shared directions along which the decision boundary is positively curved implies the existence of small universal perturbations. The existence of such shared directions is closely related to our notion of sharedness of perturbations: our notion of sharedness corresponds to common directions (perturbations) which increase classification cost for many inputs while their concept is based on shared positive curvature of the decision boundary in a direction. As discussed in "Analysis of universal adversarial perturbations", when the decision boundary is positively curved in a direction, it will lie closer to the datapoint in this direction and thus moving in this direction will increase classification cost considerably. As we have empirically shown, adversarial training is effective in reducing the sharedness of perturbations, and thus potentially also in removing shared directions in which the decision boundary is positively curved (which would be the basis for the existence of universal perturbations). While this argument is informal, we hope that the observation reported in our paper can motivate future research into a better theoretical understanding on methods for preventing universal perturbations.

---

### Official Review · AnonReviewer1 · 2017-11-28
**Work would benefit from real world tests**

**Rating:** 6
**Confidence:** 3

**Review:**

This paper analyses adversarial training and its effect on universal adversarial examples as well as standard (basic iteration) adversarial examples. It also analyses how adversarial training affects detection.

The robustness results in the paper are interesting and seem to indicate that interesting things are happening with adversarial training despite adversarial training not fixing the adversarial examples problem. The paper shows that adversarial training increases the destruction rate of adversarial examples so that it still has some value though it would be good to see if other adversarial resistance techniques show the same effect. It's also unclear from which epoch the adversarial examples were generated from in figure 5. Further the transformations in figure 5 are limited to artificially controlled situations, it would be much more interesting to see how the destruction rate changes under real-world test scenarios.

The results on the detector are not that surprising since previous work has shown that detectors can learn to classify adversarial examples and the additional finding that they can detect adversarial examples for an adversarially trained model doesn't seem surprising. There is also no analysis of what happens for adversarial examples for the detector.

Also, it's not clear from section 3.1 what inputs are used to generate the adversarial examples. Are they a random sample across the whole dataset?

Finally, the paper spends significant time on describing MaxMin and MinMax and the graphical visualizations but the paper fails to show these graphical profiles for real models.

---

> ### Author Response · Authors · 2017-12-19
> **Reply to AnonReviewer1**
>
> We would like to thank the reviewer for his comments. Regarding the reviewer's questions:
>  * Figure 5: the adversarial examples were generated from the epochs shown in the legend as "model epochs" (0, 1, 5, 51, 251). Please note that epoch 0 corresponds to a model pretrained with standard training (without adversarial training). Epoch 1 denotes the model after one additional epoch of adversarial training.
>  * Reporting how the destruction rate changes under real-world transformations would make it difficult to attribute changes in destruction rates to individual changes (since typically changes of brightness, contrast, noise would occur at the same time). Moreover, artificially controlled situations have the advantage that the amount of brightness change, blurring etc. can be systematically varied. They thus allow to systematically study in which aspects adversarial training makes a model more robust. Because of this, we focused the experiments on snythetic transformations. We would also like to note that our point in the paper is not that adversarial training makes physical world attacks impossible but rather that physical-world attacks should be tested against models hardened with, e.g., adversarial training. For instance, our results on GTSRB (see revised PDF) show that adversarial training greatly increases robustness on this dataset and also increases destruction rate of the remaining perturbations under image transformations. In the lights of these results, it would be interesting to see if the results presented in works like https://arxiv.org/abs/1707.08945v4 would carry over to a classifier hardened with adversarial training.  However, replicating this attack is beyond the scope of this paper but an important direction for future work (for which this work forms the basis).
> * In contrast to the reviewer's opinion, it was surprising for us that a detector was able to detect adversarial examples of an adversarially trained model. It was a likely assumption that detectability of perturbations were closely related to their sharedness and the existence of universal perturbations (since sharedness/universality is related to different perturbations being more "similar" and similarity in turn would make detectability easier). However, as the paper shows, adversarial training greatly reduces shared/universal perturbations but leaves detectability unchanged. Thus, these two properties seem to be unrelated which was surprising and insightful for us.
> * We do not claim that the detector could not be fooled. Our main point is not that the combination of adversarial training and detection is a robust defence but rather that adversarial training fails to become robust against certain shared patterns in adversarial perturbations that are picked up by a detector. Thus, combining adversarial training with a detection loss appears to be a promising direction for future work.
> * "Also, it's not clear from section 3.1 what inputs are used to generate the adversarial examples. Are they a random sample across the whole dataset?" If not stated otherwise, the adversarial examples were generated for the entire CIFAR10 test set. Otherwise, they are a randomly sampled subset of the test set.

---

### Official Review · AnonReviewer2 · 2017-11-29
**I like the message conveyed in this paper. However, as the statements are mostly backed by experiments, then I think it makes sense to ask how statistically significant the present results are. Moreover, is CIFAR 10 experiments conclusive enought.**

**Rating:** 6
**Confidence:** 3

**Review:**

This paper investigates the effect of adversarial training. Based on experiments using CIFAR10, the authors show that adversarial training is effective in protecting against "shared" adversarial perturbation, in particular against universal perturbation. In contrast, it is less effective to protect against singular perturbations. Then they show that singular perturbation are less robust to image transformation, meaning after image transformation those perturbations are no longer effective. Finally, they show that singular perturbations can be easily detected.

I like the message conveyed in this paper. However, as the statements are mostly backed by experiments, then I think it makes sense to ask how statistically significant the present results are. Moreover, is CIFAR 10 experiments conclusive enough.

---

> ### Author Response · Authors · 2017-12-19
> **Reply to AnonReviewer2**
>
> We would like to thank the reviewer for the comments.
>  * How statistically significant are the present results? We have run 5 repetitions of adversarial training on CIFAR10 for 50 epochs and evaluated the fooling rate of universal perturbations (as in Section 3.2). The accuracy of the model on inputs containing universal perturbations was 87.55%,  87.84%, 88.4%,  87.78%,  86.92%. As there is little variance between runs, we believe the presented results are not specific to the one run of adversarial training we investigated in more details.
>  * "Moreover, is CIFAR 10 experiments conclusive enough?" We have added an experiment on the German Traffic Sign Recognition Benchmark (GTRSB) dataset, both for the same classifier architecture used on CIFAR-10 and for a classifier using a non-residual, convolutional net. The main findings remain the same: adversarial training makes classifiers considerably more robust against universal perturbations, increases the destruction rate of perturbations considerably under most image transformations, and leaves perturbations well detectable. See the revised PDF version for more details.

---

### Public Comment · (anonymous) · 2017-11-14
**code request**

Hi, I am participating in the reproducibility challenge, would you mind sharing your code?

Thanks!

---

> ### Author Response · Authors · 2017-11-14
> **reply to code request**
>
> Thanks for your interest. At the moment, we cannot release code unfortunately. Feel free to ask any questions you have when trying to replicate our results.

---

### Decision · Program_Chairs · 2018-01-29
**ICLR 2018 Conference Acceptance Decision**

**Decision:**

Reject

**Comment:**

This paper studies to what extent adversarial training affects the properties of adversarial examples in object classification.

Reviewers found the work going in the right direction, but agreed that it needs further evidence/focus in order to constitute a significant contribution to the ICLR community. In particular, the AC encourages authors to relate their work to the growing body of (mostly concurrent) work on robust optimization and adversarial learning. For the above reasons, the AC recommends rejection at this time.